# Discrimination and Leukocyte Telomere Length by Depressive Symptomatology: The Jackson Heart Study

**DOI:** 10.3390/healthcare9060639

**Published:** 2021-05-28

**Authors:** LáShauntá M. Glover, Crystal W. Cené, Alexander Reiner, Samson Gebreab, David R. Williams, Kari E. North, Mario Sims

**Affiliations:** 1Department of Epidemiology, University of North Carolina at Chapel Hill, Chapel Hill, NC 27599, USA; kari_north@unc.edu; 2Department of Medicine, University of North Carolina at Chapel Hill, Chapel Hill, NC 27599, USA; crystal_cene@med.unc.edu; 3Department of Epidemiology, University of Washington, Seattle, WA 98195, USA; apreiner@uw.edu; 4National Institutes of Health, Bethesda, MD 20892, USA; samson.gebreab@nih.gov; 5Department of Social and Behavioral Sciences, T.H. Chan School of Public Health, Harvard University, Boston, MA 02138, USA; dwilliam@hsph.harvard.edu; 6Department of Medicine, University of Mississippi Medical Center, Jackson, MS 39216, USA; msims2@umc.edu

**Keywords:** leukocyte telomere length, discrimination, depressive symptoms, African American adults

## Abstract

Background: Psychosocial stressors, such as perceived discrimination and depressive symptoms, may shorten telomeres and exacerbate aging-related illnesses. Methods: Participants from the Jackson Heart Study at visit 1 (2000–2004) with LTL data and Center for Epidemiological Studies-Depression (CES-D) scores (*n* = 580 men, *n* = 910 women) were utilized. The dimensions of discrimination scores (everyday, lifetime, burden of lifetime, and stress from lifetime discrimination) were standardized and categorized as low, moderate, and high. Coping responses to everyday and lifetime discrimination were categorized as passive and active coping. Multivariable linear regression analyses were performed to estimate the mean difference (standard errors-SEs) in LTL by dimensions of discrimination and coping responses stratified by CES-D scores < 16 (low) and ≥ 16 (high) and sex. Covariates were age, education, waist circumference, smoking and CVD status. Results: Neither everyday nor lifetime discrimination was associated with mean differences in LTL for men or women by levels of depressive symptoms. Burden of lifetime discrimination was marginally associated with LTL among women who reported low depressive symptoms after full adjustment (b = 0.11, SE = 0.06, *p* = 0.08). Passive coping with lifetime discrimination was associated with longer LTL among men who reported low depressive symptoms after full adjustment (b = 0.18, SE = 0.09, *p* < 0.05); and active coping with lifetime discrimination was associated with longer LTL among men who reported high depressive symptoms after full adjustment (b = 1.18, SE = 0.35, *p* < 0.05). Conclusions: The intersection of perceived discrimination and depressive symptomatology may be related to LTL, and the effects may vary by sex.

## 1. Introduction

Leukocyte telomere length (LTL), or length of the ends of chromosomes in leukocyte cells, is a biomarker for cellular aging that can reflect premature declines in health at the cellular level [1]. Shorter LTL is associated with chronic diseases such as atherosclerotic cardiovascular disease, dementia, arthritis and non-cardiovascular related mortality [1,2,3]. The mechanism of how shorter LTL is associated with chronic disease is not fully understood, but studies have suggested that differences in LTL are associated with age, smoking and sedentary lifestyle [3]. Psychosocial factors, specifically chronic stressors, have recently emerged as potential risk factors for shorter telomeres and subsequent poor health [4,5]. Some studies have hypothesized that stressors experienced throughout life may cause physiological responses that promote leukocyte telomere damage [6,7]. Specifically, increased inflammatory responses and endocrine system changes from environmental stressors may directly influence activity of telomerase, causing accelerated shortening of telomeres [7]. Observational studies have found that greater self-reported stressors are associated with shorter LTL on average, even after adjusting for age and health behaviors [6,8].

Experiences of discrimination may be related to shorter LTL [9], particularly among African-American/Black adults [10,11] who report greater perceived discrimination and other psychosocial stressors [12]. These experiences and stressors contribute to racial disparities in health [13]. Structural and discriminatory barriers embedded in societal institutions within the US contribute to greater financial stress, higher reports of racial discrimination, and greater depressive symptoms among African-American adults [14]. For this reason, they may have faster shortening of telomeres when compared to White adults due to greater exposure to chronic stress [15,16,17]. Research on the association of race/ethnicity and LTL has been mixed. While some reports have found that African-American adults have longer telomeres than White adults [7,18], other work has shown that African-American persons have faster shortening of telomeres than White persons [19,20]. Despite these mixed results, there is an apparent link between psychosocial stress and shorter LTL among African-American adults [8,10,16,21], and there are noted sex differences [8,21]. Depression, which is also highly prevalent among those who report greater psychosocial stress, is also inversely associated with shorter LTL [22]. Though a greater prevalence of depression is reported among White adults, there is a disparity in chronic depression, more severe symptoms of depression, and treatment of depression among African-American adults [23,24]. Little is known about the intersection of perceived discrimination and depressive symptoms on LTL among African-American men and women.

In this study of men and women from the Jackson Heart Study (JHS), we explored the association between multiple dimensions of discrimination (everyday, lifetime, burden of lifetime, and stress from lifetime discrimination) and coping with every day and lifetime discrimination and LTL by depressive symptomatology. This approach enables us to better understand the intersectionality of discrimination and depressive symptomatology and its relation to immune health among men and women. We hypothesized an inverse association between dimensions of discrimination and LTL by greater depressive symptoms, with differences by sex. We also hypothesized that passive and active coping will be positively and negatively associated with LTL, respectively, with greater depressive symptoms, and that there will be also be differences by sex.

## 2. Materials and Methods

### 2.1. Participants

The JHS is a prospective cohort study of CVD among adults who self-identified as African-American and were from the Jackson, MS metropolitan area (3371 women, 1935 men; age 20–95 years old). The baseline exam (visit 1) occurred from 2000 to 2004. Eligible participants were non-institutionalized and were either from the Atherosclerosis Risk in Communities (ARIC) Study (30%), the Mississippi Department of Transportation Driver’s License and Identification List (17%), volunteers (22%), or family members of those who had already agreed to be a part of the study (31%). Demographic, socioeconomic, psychosocial, health history, and clinical data were collected from completed in-home interviews, self-administered questionnaires, and in-clinic examinations at each visit, including exam 2 (2005–2008) and exam 3 (2009–2013). Additional information about study design, recruiters, and incentives can be found at Fuqua et al. [25] All JHS participants provided written informed consent and all study protocols conform to the 1975 Declaration of Helsinki guidelines. The JHS was also approved by the Institutional Review Boards of participating institutions: The University of Mississippi Medical Center, Jackson State University, and Tougaloo College.

### 2.2. Leukocyte Telomere Length

During exam 1, a subset of participants were a part of an ancillary study which collected DNA samples to extract leukocyte telomere length data (*n* = 2573). The ancillary study sample was drawn from the family sample of participants who consented for future genetic analysis and were not a part of the ARIC study. DNA was extracted from whole blood using Puregene reagents (Gentra Systems Inc, Minneapolis, MN 55441, USA) [26], and the Southern blot analysis method was used to measure LTL in kilobases (kb) at the Center of Human Development and Aging at the Rutgers New Jersey Medical School [27]. For quality control, the assessment of DNA integrity was conducted prior to LTL measurement. The average inter-assay coefficient of variation was 2.0% and the interclass correlation coefficient for individual measures of LTL was 0.95. Participants with missing LTL values or those who had inadequate DNA content were removed, leaving an analytic sample of *n* = 2518 (63.5% women). Missing values were more common among men but there were no significant differences in age when compared to those who were not missing LTL data. The distribution of LTL is normal in the JHS.

### 2.3. Measures of Perceived Discrimination

The main exposures include four dimensions of perceived discrimination (everyday, lifetime, burden from lifetime, and stress from lifetime discrimination), which were used to capture discriminatory events across multiple circumstances. In addition to measures of discrimination, we also analyzed coping responses to everyday and lifetime discrimination.

Everyday discrimination was adapted from the Williams’ scale [28], and had good internal reliability (α = 0.88) in JHS. Participants were asked “How often on a day-to-day basis do you have the following experiences?”: “You are treated with less courtesy…, You are treated with less respect…, You receive poorer service than others at restaurants…, People act as if they think you are not smart…, People act as if they are afraid of you…, People act as if they think you are dishonest…, People act as if they think you are not as good as they are…, You are called names or insulted…, You are threatened.” Responses ranged from 1 (“never”) to 7 (“several times a day”).

Lifetime discrimination was based on the scale developed by Krieger and Sydney [29] and had an internal reliability (α = 0.78) that is consistent with that of major life event scales. Participants were asked about the occurrence of unfair treatment over the lifetime (yes/no) across nine domains: at school, getting a job, at work, getting housing, getting resources/money, getting medical care, on the street or public place, getting services or other ways. The count of the domains (0–9) created the lifetime discrimination score.

Burden of lifetime discrimination was measured by asking the following among those who reported at least 1 experience of lifetime discrimination: “Overall, how much has discrimination interfered with you having a full productive life” and “Overall, how much harder has your life been because of discrimination?... not at all, a little, some, or a lot?” Responses were summed to create a score ranging from 1 to 4, where higher scores indicated greater burden. The internal reliability (0.63) was acceptable given that each question measured different constructs.

Stress from lifetime discrimination was also restricted to persons who reported at least 1 instance of lifetime discrimination. Because previous research suggested that the degree of stress was a key determinant of the adverse health consequences of discrimination, the “stress” component of the “burden” index was examined separately. Participants were asked, “When you had experiences like these over your lifetime, have they been not stressful, moderately stressful, or very stressful?” The response to this one question was used to capture stress from lifetime discrimination.

### 2.4. Coping with Discrimination

Passive and active coping responses to everyday and lifetime discrimination were measured from the following prompt “And when you receive lesser or unfair treatment in your day-to-day life, do you usually/thinking back to experiences over your lifetime…what did you do”. Responses such as “Speak up”, “Try to change it”, “Work harder to prove them wrong” “Get violent” were classified as active coping responses, while “Accept it”, “Ignore it”, “Keep it to yourself”, “Pray”, “Avoid it”, “Forget it” were classified as passive coping responses [30]. “Other” was also an option that was not classified as passive or active coping, and was used separately for comparison.

The dimensions of discrimination scores were transformed into standard deviation units and categorized into low, moderate, and high based on a tertile distribution. Coping responses were grouped into binary categories (passive (vs. active and other) and active (vs. passive and other)) based on the response option participants chose.

### 2.5. Depressive Symptoms

We included depressive symptoms as a modifier in these analyses. The Centers for Epidemiological Studies (CES-D) scale was used to capture depressive symptoms in the JHS. The CES-D was developed to capture the presence or absence of clinically significant depressive symptoms over a one week period. The scale has 20 items, with a range of 0–60. Binary classification is recommended to identify individuals at risk for clinical depression, where a score < 16 indicates low risk and a score ≥ 16 indicates high risk [31,32].

### 2.6. Covariates

We adjusted for age, educational attainment, waist circumference [33], smoking [34], and CVD prevalence [35] to reduce confounding bias in our analyses. Age was continuous and was collected in years. The categories for educational attainment include < High School diploma (referent), High School Diploma to Some College/Trade School, and College Degree or more. Waist circumference in centimeters was used to capture obesity. We used categories from the American Heart Association’s Life Simple 7 categorized as poor (referent) vs. ideal for smoking, where individuals who were current smokers or had quit smoking less than 12 months ago at baseline were categorized as having ideal smoking health and individuals who identified as current smokers at baseline were categorized as having poor smoking health. CVD history (yes/no) was determined by self-reported physician diagnosis of heart attack, angina, or stroke, or self-reported history of heart procedures such as coronary bypass or heart catheterization. Electrocardiogram measurements at exam 1 were also used to determine CVD history.

### 2.7. Statistical Analysis

Of the 2518 participants with LTL data, 1526 participants had complete depressive symptoms data. When comparing participants who had complete depressive symptoms data to those who did not, there were significant differences by mean age, educational attainment and mean LTL. Specifically, participants with depressive symptoms data were younger (53.2 years vs. 57.13 years, *p* < 0.001), more educated (College degree or more 44.9% vs. 30.13%, *p* < 0.001), and had longer LTL (7.25 kb vs. 7.07 kb, *p* < 0.001) (data not shown). We removed those with missing covariates: education (*n* = 5), smoking (*n* = 18), waist circumference (*n* = 5), CVD (*n* = 15). The final analytic sample was 1490. Characteristics of the sample by sex (men = 580, women = 910) were tested using chi square and Kruskal–Wallis tests. To examine moderation by sex and depressive symptoms, we examined the distribution of measures of discrimination and LTL by sex and depressive symptoms and utilized figures [36]. We presented the predicted LTL by burden of and stress from lifetime discrimination and depressive symptoms among men and women in a figure to graphically demonstrate how associations of discrimination and LTL were moderated by depressive symptoms. Multivariable linear regression analyses were performed to estimate the mean difference (standard errors—SEs) in LTL by each measure of discrimination stratified by depressive symptoms among men and women. We used *p* value for trend (*p* for trend) estimates to represent the linear trend in the association with LTL across discrimination categories. We also used multivariable linear regression analyses to estimate the mean difference in LTL by coping responses to discrimination stratified by depressive symptoms among men and women. Model 1 adjusted for age and educational attainment, and model 2 adjusted for model 1 + waist circumference, smoking, and prevalent CVD. A *p*-value < 0.05 was considered statistically significant. All analyses were conducted in Stata 13.1 (Stata Corp, College Station, TX, USA).

## 3. Results

### 3.1. Sample Characteristics

Table 1 presents the sample characteristics by sex. The mean LTL for men was significantly shorter than women (7.14 kb vs. 7.32 kb, *p* < 0.001). Women were more likely to have ideal smoking and report greater depressive symptoms than men (*p* < 0.01). Women also reported higher percentages of passive and active coping with everyday discrimination than men (*p* < 0.001). Men reported a higher burden of lifetime discrimination than women (*p* < 0.05).

### 3.2. Discrimination and LTL by Sex and Depressive Symptoms

Table 2 presents the associations of dimensions of perceived discrimination with mean differences in LTL stratified by levels of depressive symptoms for men and women. High (vs. low) everyday discrimination was not associated with mean LTL for men or women who reported CES-D scores < 16 or ≥ 16 in minimally adjusted and fully-adjusted models. Everyday discrimination, as measured in SD units, was also not associated with LTL. Lifetime discrimination was not associated with LTL for men or women by levels of CES-D scores. Burden of lifetime discrimination was marginally associated with LTL among women who reported CES-D scores < 16 after full adjustment (b = 0.11, SE = 0.06, *p* = 0.08), but the P for trend value was not significant (*p* < 0.05) Each 1-SD increase in burden of lifetime discrimination was associated with an increased mean difference in LTL among women who reported CES-D score < 16 after adjustment for age and educational attainment (b = 0.05, SE = 0.03, *p* < 0.05). Moderate (vs. low) stress from lifetime discrimination was marginally associated with mean LTL among men who reported low depressive symptoms (b = −0.14, SE = 0.07, *p* = 0.05) after adjustment for covariates in models 1 and 2. *p* for trend estimates were marginally significant (*p* < 0.10). These findings were not significant for women, though estimates were in the hypothesized direction. For women with low depressive symptoms, there was a stronger dose-response effect for stress from lifetime discrimination categories in models 1 and 2 (*p* for trend < 0.05).

### 3.3. Moderation by Depressive Symptoms

We demonstrated evidence of moderation by depressive symptoms in the association of burden of and stress from lifetime discrimination with LTL in both men and women (Figure 1). Among men with high (vs. low) depressive symptoms, mean LTL was shorter for those who reported low stress from lifetime discrimination (panel B). Among women with high (vs. low) depressive symptoms, the mean LTL was longer for those who experienced low burden from lifetime discrimination and high stress from lifetime discrimination (panels C and D).

### 3.4. Coping with Discrimination and LTL by Sex and Depressive Symptoms

Table 3 presents the associations of passive and active coping with everyday and lifetime discrimination with LTL by CES-D scores among men and women. There was no association between passive or active coping with everyday discrimination and LTL among men regardless of CES-D score. Although men who reported active coping with everyday discrimination had shorter mean LTL, findings were non-significant. Passive (vs. active and other) coping with lifetime discrimination was significantly and positively associated with LTL among men who reported low CES-D scores after full adjustment (b = 0.18, SE = 0.09, *p* < 0.05). Active (vs. passive and other) coping with lifetime discrimination was positively and significantly associated with LTL among men who reported CES-D scores ≥ 16 in the fully adjusted model (b = 1.18, SE = 0.35, *p* < 0.05). Coping with everyday discrimination was not significantly associated with LTL among women with low or high CES-D scores. Estimates were not significant for passive and active coping with lifetime discrimination by CES-D score, though the estimates were in the hypothesized direction.

## 4. Discussion

We investigated the associations of multiple measures of perceived discrimination and coping responses with LTL by depressive symptomatology among men and women. We found evidence of moderation of depressive symptoms in the association of perceived discrimination with LTL among men and women. In fully-adjusted models, men had shorter LTL if they reported moderate stress from lifetime discrimination and low depressive symptoms. Moderate burden of lifetime discrimination was associated with longer LTL among women who reported low depressive symptoms after full adjustment. We also found evidence of moderation by depressive symptoms in the association of coping responses to discrimination with LTL. Men who reported passive and active coping responses to lifetime discrimination had longer LTL, for both high and low levels of depressive symptoms. Overall, our study found sex differences in the associations of discrimination and coping with discrimination with LTL by depressive symptomatology among African Americans.

Previous research has found differences in LTL by discrimination type and race/ethnicity. Liu et al. [37] examined associations of lifetime and everyday discrimination with LTL among African-American and White adults in the 2008 Health and Retirement Study (HRS), and found that everyday discrimination was inversely associated with LTL in African-American individuals, but not among White individuals. Other research found no significant association of depressive symptoms and LTL [18,22]; however, a meta-analysis of studies examining clinical depression and LTL found shorter LTL among depressed individuals [22] and there are noted differences by sex and antidepressant use [38]. In the JHS, there was no cross-sectional association between depressive symptoms and LTL [21]. However, there are few studies that have analyzed the association of discrimination and LTL by depressive symptomatology. Chae et al. [8] conducted a cross-sectional study of racial discrimination and LTL among 92 African American men from the San Francisco Bay area. They reported evidence of moderation by depressive symptoms and found that racial discrimination was associated with shorter LTL among men with low depressive symptoms. Our study found this association for moderate burden of lifetime discrimination among men. In addition, concordant with Chae et al. [8], we found no significant main effect for measures of perceived discrimination and LTL nor depressive symptoms and LTL among men nor women (data not shown). Our analyses extended their study by including multiple measures of discrimination, coping strategies for everyday and lifetime discrimination, a large sample of women, and a larger sample of African-American men.

Though the discrimination results for men were similar to Chae et al. [8], we found unexpected associations for women. Among women with moderate burden of lifetime discrimination and low depressive symptoms, there was a marginally positive association with LTL. Women in this sample had longer LTL than men. It is possible that women with lower depressive symptoms were more likely to have longer LTL. Studies have documented longer LTL in women [39,40], which suggests women may be healthier or have greater survival than men [41,42]. Others explain longer LTL as having a greater propensity for shortening because longer telomeres are more susceptible to free radicals [43]. For both men and women with high depressive symptoms, non-significant associations were mostly in the positive direction for most measures of discrimination and LTL, which suggests that experiences of discrimination may not shorten LTL, even for those who are also at higher risk of depression.

We hypothesized that active coping with discrimination would be associated with shorter LTL among men and women with both high and low depressive symptoms. We expected passive coping with discrimination would be associated with longer LTL for those with both low and high depressive symptoms. We found longer LTL among men with low depressive symptoms who used passive coping responses to lifetime discrimination. In addition, we found that active coping with lifetime discrimination was significantly associated with longer LTL among men with high depressive symptoms. This finding suggests that both coping responses to lifetime discrimination may be associated with longer LTL for men, and one coping strategy may be more influential than the other depending on depressive symptomatology. Though little is known about coping responses to discrimination and biological aging, avoidance and spirituality are reported as effective coping strategies among African-American men and women [44]. Within JHS, men tend to use more active coping and are less religious than women [45]. Aggressive coping types such as anger and resentment are common coping strategies among men, but are associated with cognitive discomfort and poor mental health [46]. Related active coping strategies, such as use of John Henryism, has differing health implications. For instance, while some have reported negative health effects for high John Henryism and health [47], some studies have found beneficial health effects among African-American men with higher scores on the John Henryism scale [48,49], which may align with our study. Further research is needed to confirm how coping strategies influence LTL while experiencing psychosocial stress among African-American individuals, especially men. Resilience mechanisms for women may also be important for biological aging and should also be considered in future research.

Limitations of this study include use of an all African-American sample from MS, which limits generalizability to African-American adults in other areas of the US. Also, we did not examine attribution or perceived reasons for discrimination, such as race, sex, height or weight. The cross-sectional design precludes drawing causal inferences and determining directionality of associations. A prospective design would allow for a more dynamic assessment of associations of discrimination with changes in LTL over time. These analyses were also not corrected for multiple testing, and associations should be considered hypothesis generating and should be interpreted accordingly. It is possible that covariates included could have dual roles as mediators and confounders, which was difficult to disentangle in the analyses. Also, LTL was measured among related individuals and is highly correlated with genetic ancestry, and analyses did not account for relatedness. Though there were many limitations, the strengths of this study include use of a large, socioeconomically heterogeneous sample of African-American adults. This study also includes multiple well-characterized measures of discrimination including everyday, lifetime, burden of lifetime discrimination and coping responses to everyday and lifetime discrimination. Additionally, we examined the intersectionality of discrimination and depressive symptoms among African-American adults in order to better understand the cumulative burden of social determinants on LTL among this at-risk subpopulation.

In conclusion, we found marginal positive and negative associations between burden and stress that result from lifetime discrimination and LTL for men and women with low depressive symptoms, respectively. We also found that passive and active coping strategies are related to longer LTL for men regardless of levels of depressive symptoms. Given the racial disparities in physical and mental health among African-American adults, more studies should investigate the intersection of discrimination and depressive symptoms and its relation to biomarkers of cellular aging that are associated with downstream CVD risk and mortality. Additional studies will also inform social and health policy by considering structural forms of discrimination that impact mental and physical health.

## Figures and Tables

**Figure 1 healthcare-09-00639-f001:**
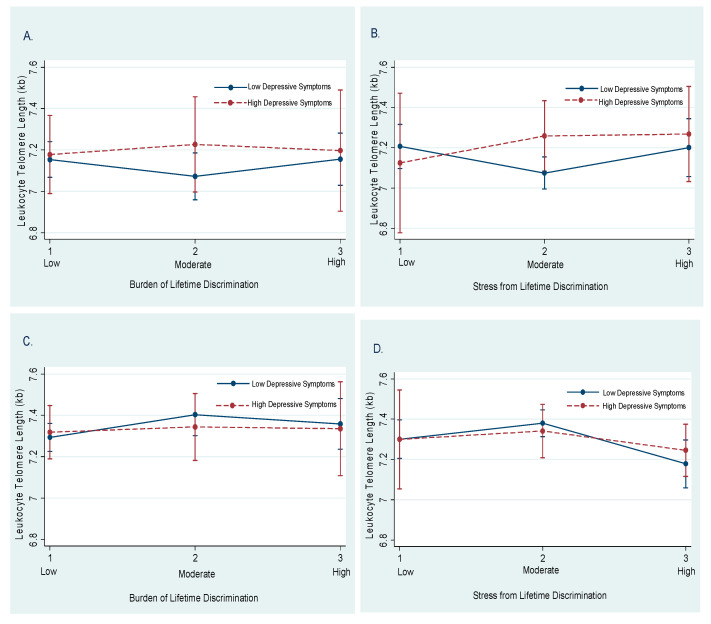
Predicted associations of leukocyte telomere length (LTL) with burden of and stress from lifetime discrimination by depressive symptoms among men (**A**,**B**) and women (**C**,**D**) with 95% confidence intervals (Cis), Jackson Heart Study (2000–2004). Note: Models were fully-adjusted for the association of mean differences in LTL at baseline with low, moderate, and high burden of and stress from lifetime discrimination by depressive symptoms. Depressive symptoms were categorized as low (Centers for Epidemiologic Studies-Depression (CES-D) > 16) and high (CES-D score ≤ 16). All predicted models are adjusted for age, education, waist circumference, smoking, and CVD prevalence.

**Table 1 healthcare-09-00639-t001:** Select characteristics by sex, Jackson Heart Study (2000–2004).

Characteristics	Men (*n* = 580)	Women (*n* = 910)	*p*-Value
Age (mean ± SD)	52.46 ± 12.34	53.51 ± 11.73	0.101
Education (%)			0.480
<HS	12.59	14.30	
HS–Some College	40.52	41.58	
College Degree or More	46.90	44.11	
Waist circumference (mean ± SD)	101.39 ± 15.61	100.89 ± 16.79	0.564
Smoking (%)			0.007
Poor Smoking Health	17.93	12.86	
Ideal Smoking Health	82.07	87.14	
Mean LTL (mean ± SD)	7.14 ± 0.70	7.32 ± 0.67	<0.001
Mean CES-D score (mean ± SD)	10.08 ± 7.38	11.98 ± 8.55	<0.001
Everyday Discrimination (%)			0.415
Low	46.82	50.48	
Moderate	26.97	24.94	
High	26.22	24.58	
Lifetime Discrimination (%)			0.079
Low	39.92	46.07	
Moderate	33.52	29.52	
High	26.55	24.40	
Burden of Lifetime Discrimination (%)			0.035
Low	49.22	55.67	
Moderate	28.68	27.15	
High	22.09	17.19	
Stress from Lifetime Discrimination (%)			0.081
Low	23.03	26.01	
Moderate	52.86	54.80	
High	24.11	19.19	
Coping with Everyday Discrimination (%)			0.001
Passive	45.06	54.35	
Active	43.80	52.47	
Coping with Lifetime Discrimination (%)			0.173
Passive	14.57	17.46	
Active	6.19	5.01	

Abbreviations: HS, High School; SD, Standard Deviation. *p* values are from Chi Square and Kruskal–Wallis tests.

**Table 2 healthcare-09-00639-t002:** Mean difference (standard errors) in leukocyte telomere length by perceived discrimination and depressive symptoms among men and women, Jackson Heart Study (2000–2004).

	MEN				WOMEN			
	CES-D Score < 16 (*n* = 220)		CES-D Score ≥ 16 (*n* = 99)		CES-D Score < 16 (*n* = 690)		CES-D Score ≥16 (*n* = 220)	
	Model 1	Model 2	Model 1	Model 2	Model 1	Model 2	Model 1	Model 2
Everyday								
Low (referent)	1.00		1.00		1.00		1.00	
Moderate	0.02 (0.07)	0.02 (0.07)	0.07 (0.16)	0.07 (0.17)	−0.02 (0.06)	−0.02 (0.06)	0.15 (0.10)	0.16 (0.10)
High	−0.08 (0.08)	−0.07 (0.08)	0.05 (0.05)	0.08 (0.15)	−0.02 (0.06)	−0.01 (0.06)	−0.10 (0.11)	−0.10 (0.12)
*p* for trend	0.424	0.500	0.896	0.840	0.985	0.940	0.136	0.108
SD units	−0.02 (0.03)	−0.01 (0.03)	−0.01 (0.05)	−0.004 (0.05)	−0.006 (0.03)	−0.01 (0.03)	−0.001 (0.04)	−0.004 (0.05)
Lifetime								
Low (referent)	1.00		1.00		1.00		1.00	
Moderate	−0.06 (0.07)	−0.05 (0.07)	−0.07 (0.14)	−0.08 (0.14)	−0.08 (0.06)	−0.08 (0.06)	0.05 (0.10)	0.06 (0.11)
High	0.004 (0.08)	0.02 (0.08)	−0.05 (0.16)	−0.05 (0.17)	0.04 (0.07)	0.04 (0.07)	0.09 (0.10)	0.10 (0.11)
*p* for trend	0.702	0.686	0.883	0.858	0.177	0.252	0.731	0.637
SD units	−0.01 (0.03)	−0.01 (0.03)	−0.04 (0.07)	−0.03 (0.07)	0.01 (0.03)	0.01 (0.03)	0.02 (0.04)	0.02 (0.04)
Burden of Lifetime								
Low (referent)	1.00		1.00		1.00		1.00	
Moderate	−0.10 (0.07)	−0.07 (0.07)	0.05 (0.14)	0.06 (0.15)	**0.11 (0.06) ^‡^**	**0.11 (0.06) ^‡^**	0.01 (0.10)	0.03 (0.10)
High	−0.02 (0.08)	0.006 (0.08)	0.02 (0.17)	0.01 (0.17)	0.07 (0.07)	0.06 (0.07)	0.02 (0.13)	0.12 (0.13)
*p* for trend	0.464	0.538	0.850	0.929	0.128	0.200	0.994	0.965
SD units	0.01 (0.03)	0.01 (0.03)	0.06 (0.06)	0.06 (0.06)	**0.05 (0.03) ^‡^**	0.04 (0.03)	−0.001 (0.05)	0.01 (0.05)
Stress from Lifetime								
Low (referent)	1.00		1.00		1.00		1.00	
Moderate	**−0.14 (0.07) ^‡^**	**−0.14 (0.07) ^‡^**	0.16 (0.19)	0.23 (0.19)	0.07 (0.06)	0.08 (0.06)	0.04 (0.13)	0.04 (0.13)
High	0.01 (0.09)	−0.01 (0.09)	0.07 (0.20)	0.14 (0.20)	−0.12 (0.08)	−0.12 (0.08)	−0.06 (0.13)	−0.06 (0.13)
*p* for trend	0.076	0.092	0.698	0.454	**0.027 ***	**0.015 ***	0.615	0.557
SD units	−0.04 (0.03)	−0.03 (0.03)	−0.02 (0.05)	−0.02 (0.05)	0.01 (0.03)	0.01 (0.03)	0.001 (0.04)	0.001 (0.04)

Abbreviations: CES-D, Centers for Epidemiological Studies; SD, Standard Deviation. Model 1 adjusted for age and educational attainment. Model 2 adjusted for model 1 + waist circumference, smoking, and CVD prevalence. Bold—^‡^
*p* < 0.1 and * *p* < 0.05.

**Table 3 healthcare-09-00639-t003:** Mean differences (standard errors) in leukocyte telomere length by coping with perceived discrimination and depressive symptoms among men and women, Jackson Heart Study (2000–2004).

	MEN				WOMEN			
	CES-D Score < 16 (*n* = 220)		CES-D Score ≥ 16 (*n* = 99)		CES-D Score < 16 (*n* = 690)		CES-D Score ≥ 16 (*n* = 220)	
	Model 1	Model 2	Model 1	Model 2	Model 1	Model 2	Model 1	Model 2
Everyday								
Passive vs. Active & Other	0.05 (0.07)	0.05 (0.07)	0.10 (0.12)	0.03 (0.13)	0.003 (0.05)	−0.02 (0.05)	−0.11 (0.09)	−0.12 (0.09)
Active vs. Passive & Other	−0.09 (0.06)	−0.10 (0.06)	−0.14 (0.12)	−0.07 (0.13)	−0.01 (0.05)	−0.001 (0.05)	0.12 (0.09)	0.13 (0.09)
Lifetime								
Passive vs. Active & Other	**0.17 (0.09) ^‡^**	**0.18 (0.09) ***	0.08 (0.18)	0.03 (0.18)	−0.005 (0.07)	−0.01 (0.07)	−0.01 (0.10)	−0.02 (0.10)
Active Vs. Passive & Other	−0.14 (0.13)	−0.13 (0.13)	**0.95 (0.33) ***	**1.18 (0.35) ***	−0.02 (0.12)	−0.05 (0.12)	−0.04 (0.23)	−0.04 (0.23)

Abbreviations: CES-D, Centers for Epidemiological Studies. Model 1 adjusted for age and educational attainment. Model 2 adjusted for model 1 + waist circumference, smoking, and CVD prevalence. Bold—^‡^
*p* < 0.1 and * *p* < 0.05.

## Data Availability

The data used for this study can be requested for purposes of reproducing results. Request to access this data set (or other data in the JHS) may be directed to the qualified researchers trained in human subject confidentiality within the JHS Coordinating Center atjhsccrc@umc.edu.

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
