# Peer review of "Discrimination and Leukocyte Telomere Length by Depressive Symptomatology: The Jackson Heart Study"

_healthcare, 2021, doi:10.3390/healthcare9060639_

Round 1

Reviewer 1 Report

The role of telomeres in the stability and the integrity of the genome is well documented. Telomere shortening has been demonstrated in patients with cancer or inflammatory diseases. In this study, Glover et al. analyzed telomere length in a large cohort from Jackson Heart Study using TRF assay. Significant difference between telomere length of men and women has been showed. The authors demonstrated a significant association between burden and stress that results from lifetime discrimination and telomere length for men and women with low Healthcare depressive symptoms.  

Major remarks: 

  1. The results need to be simplified and clarified. Paragraphs can make the results easier to follow 
  2. In the discussion, the authors listed some limitation of this study. However, it will be important to validate these data with the use of iterative samples and sequential telomere length quantification should be performed. The change of telomere length can be a better biomarker of poor clinical outcome of the subset of these patients. 
  3. It is important to discuss the relevance of this biomarker for therapeutic targets, as well as for the clinical outcomes of these patients. 

Author Response

Reviewer 1 

Major remarks: 

Comment: The results need to be simplified and clarified. Paragraphs can make the results easier to follow 

            Response: We have added subtitles within the results section to make the results easier to follow. (See Results)

Comment: In the discussion, the authors listed some limitation of this study. However, it will be important to validate these data with the use of iterative samples and sequential telomere length quantification should be performed. The change of telomere length can be a better biomarker of poor clinical outcome of the subset of these patients. 

            Response: Thank you for this comment. We agree that this is a limitation of the study as we have mentioned in the discussion section (See Discussion, paragraph 5, lines 143-145). At this juncture, it is impossible to validate these data as they are not available. Additionally, LTL measures at multiple time points are also unavailable. In the discussion, we also compared our study to other studies that have examined similar associations as a way to validate our results.

Comment: It is important to discuss the relevance of this biomarker for therapeutic targets, as well as for the clinical outcomes of these patients. 

            Response: As far as we know, LTL is not relevant as a biomarker for therapeutic target, rather it is a marker of cellular aging. If we are knowledgeable of biomarkers that can reflect premature declines in health, then we would understand the mechanisms by which disease occurs. This was also stated in the first paragraph of the introduction.

Reviewer 2 Report

Abstract and Introduction: The introduction of the paper has been well-written.It summarizes the current research in the area as well as makes a good case for the present study. It would be helpful to mention other research studies done previously that have looked at the association between experiences of discrimination, coping and other physiological measures of stress.

Methodology: Please provider more details regarding the methods. The study data has been collected with a combination of interviews and questionnaires, however no details have been provided regarding the interview questions or interviewers who conducted those in-home interviews. It is also unclear what kind of information was collected using interviews. Please mention if the participants received any form of compensation for their participation in the study.

Results: The results section of the paper is good, especially tables and figure make it easy to understand the findings and give this section a nice flow.

Discussion: It would be helpful to add a paragraph on the significance of these findings for clinicians as well as for policy makers.

It is a well done research and paper has been written well overall. The paper can be reconsidered after suggested changes have been made in the Methodology and discussion section.

Author Response

Reviewer 2

Comment: Abstract and Introduction: The introduction of the paper has been well-written. It summarizes the current research in the area as well as makes a good case for the present study. It would be helpful to mention other research studies done previously that have looked at the association between experiences of discrimination, coping and other physiological measures of stress.

            Response: Thank you. Previous studies have not brought together/ examined the intersection of discrimination, coping, and LTL. These associations have been examined separately and were mentioned in the introduction.

Comment: Methodology: Please provider more details regarding the methods. The study data has been collected with a combination of interviews and questionnaires, however no details have been provided regarding the interview questions or interviewers who conducted those in-home interviews. It is also unclear what kind of information was collected using interviews. Please mention if the participants received any form of compensation for their participation in the study.

            Response: Interview questions related to this project was mentioned in other sections of the methods (See Measures of Perceived Discrimination, Coping with Discrimination, Depressive Symptoms, and Covariates). We added a sentence in the first paragraph of the methods that references a paper that discusses the study design, recruiters, and incentives relevant to the cohort study overall (Fuqua et al. 2005).

Fuqua, S.R., et al., Recruiting African-American research participation in the Jackson Heart Study: methods, response rates, and sample description. Ethn Dis, 2005. 15(4 Suppl 6): p. S6-18-29.

Comment: Results: The results section of the paper is good, especially tables and figure make it easy to understand the findings and give this section a nice flow.

            Response: Thank you.

Comment: Discussion: It would be helpful to add a paragraph on the significance of these findings for clinicians as well as for policy makers.

            Response: Because there are few papers that have investigated the intersectionality of experiences of discrimination, depressive symptoms, and leukocyte telomere length, we feel there is not enough evidence to discuss findings for clinicians and policy makers. We believe additional studies will lead to informing social and health policy. We added a sentence to the last paragraph of the manuscript

Comment: It is a well done research and paper has been written well overall. The paper can be reconsidered after suggested changes have been made in the Methodology and discussion section.

            Response: Thank you.